# Development and application of species ID and insecticide resistance assays, for monitoring sand fly *Leishmania* vectors in the Mediterranean basin and in the Middle East

Sofia Balaska[1,2]\*, Jahangir Khajehali[3], Konstantinos Mavridis[1], Mustafa Akiner[4], Kyriaki Maria Papapostolou[1], Latifa Remadi[1], Ilias Kioulos[5], Michail Miaoulis[6], Emmanouil Alexandros Fotakis[1,7], Alexandra Chaskopoulou[6], John Vontas[1,5]\*

1 Institute of Molecular Biology & Biotechnology, Foundation for Research & Technology Hellas, Heraklion, Greece, 2 Department of Biology, University of Crete, Heraklion, Greece, 3 Department of Plant Protection, College of Agriculture, Isfahan University of Technology, Isfahan, Iran, 4 Recep Tayyip Erdogan University, Department of Biology, Zoology Section, Rize, Turkey, 5 Department of Crop Science, Agricultural University of Athens, Athens, Greece, 6 European Biological Control Laboratory, USDA-ARS, Thessaloniki, Greece, 7 Department of Infectious Diseases, Istituto Superiore di Sanità, Rome, Italy

\* sofia_balaska@imbb.forth.gr (SB); vontas@imbb.forth.gr (JV)

## Abstract

### Background

Development of insecticide resistance (IR) in sand fly populations is an issue of public health concern, threatening leishmaniasis mitigation efforts by insecticide-based vector control. There is a major knowledge gap in the IR status of wild populations worldwide, possibly attributed to the unavailability of specialized tools, such as bioassay protocols, species baseline susceptibility to insecticides and molecular markers, to monitor such phenomena in sand flies.

### Methodology/Principal findings

Sand fly populations from (semi-)rural regions of Greece, Turkey and Iran were sampled and identified to species, showing populations' structure in accordance with previously reported data. Genotyping of known pyrethroid resistance-associated loci revealed the occurrence of voltage-gated sodium channel (*vgsc*) mutations in all surveyed countries. Knock-down resistance (*kdr*) mutation L1014F was prevalent in Turkish regions and L1014F and L1014S were recorded for the first time in Iran, and in Turkey and Greece, respectively, yet in low frequencies. Moreover, CDC bottle bioassays against pyrethroids in mixed species populations from Greece indicated full susceptibility, using though the mosquito discriminating doses. In parallel, we established a novel individual bioassay protocol and applied it comparatively among distinct *Phlebotomus* species' populations, to detect any possible divergent species-specific response to insecticides. Indeed, a significantly different knock-down rate between *P. simici* and *P. perfiliewi* was observed upon exposure to deltamethrin.

**Data Availability Statement:** All relevant data are within the paper and its Supporting information files.

**Funding:** This work was financially supported by the Hellenic Foundation for Research and Innovation (HFRI) under the 3rd Call for HFRI PhD Fellowships awarded to S.B. (Fellowship Number: 11078), by Fondation Santé through a research grant awarded to K.M. (Grant number: ΔΕΘ000155), while L.R. was supported by a Marie Curie Post Doctoral Fellowship (HORIZON-MSCA-2023-PF-01-01, project: 101152599). Additionally, partial financial support was received by the European Union's Horizon Europe research and innovation programme under grant agreement number 101046133 (ISIDORe). The funders had no role in study design, data collection and analysis, decision to publish, or preparation of the manuscript.

**Competing interests:** The authors have declared that no competing interests exist.

## Conclusions/Significance

IR in sand flies is increasingly reported in leishmaniasis endemic regions, highlighting the necessity to generate additional monitoring tools, that could be implemented in relevant eco-epidemiological settings, in the context of IR management. Our molecular and phenotypic data add to the IR map in an area with otherwise limited data coverage.

## Author summary

Phlebotomine sand flies transmit leishmaniasis to humans and animals, a neglected tropical disease of the (sub-)tropics, currently expanding though in previously non-endemic regions. As leishmaniasis eradication largely relies on vectors' insecticidal control, regular monitoring of insecticide resistance (IR) is a core element of integrated vector management. IR data are limited in sand fly populations worldwide though, due to the unavailability of robust molecular diagnostic and phenotyping tools, as well as deficiencies in human resources, appropriate training and capacity building. Here, we aim to characterise the pyrethroid resistance status of populations originating from countries of the Mediterranean basin and the Middle East; i.e. Greece, Turkey and Iran. Pyrethroid resistance-associated mutations were detected in all three countries, while the populations from Greece exhibited susceptibility upon exposure to deltamethrin in CDC bioassays. We established an individual bioassay protocol to enable IR monitoring in settings with multi-species population structure, and applied it comparatively among three distinct *Phlebotomus* species, revealing that they respond differently to insecticide treatment. Our study contributes data for sustainable evidence-based sand fly control, and highlights the importance of systematic IR surveillance programs in leishmaniasis endemic regions.

## Introduction

Leishmaniasis is classified among the top ten epidemiologically impacting neglected tropical diseases, causing approximately 30,000 parasite-causative fatalities out of almost one million cases per year, globally [1]. Phlebotomine sand flies mediate the transmission of *Leishmania* parasites to humans and animals, in the tropical and sub-tropical zones around the world [1].

Even though low- and middle-income countries are disproportionately burdened by leishmaniases occurrence [1, 2], the epidemiological scenarios in central and southern Europe and the Mediterranean basin have been changing, over the last decades, with new disease foci appearing in previously non-endemic regions [2–4]. Some key determinants accounting for this changing epidemiological trend are the climatic crisis along with environmental modifications/urbanization, facilitating the spread of sand flies northwards, and occasionally in urban settings, as well as the increasing population movement (i.e. migration/ mass population displacement, tourism, travelling with reservoir hosts, such as dogs) [5,6].

The role of vector control to combat leishmaniasis transmission is crucial, especially due to the unavailability of vaccines for humans, as well as several issues accompanying antileishmanial chemotherapy (e.g. drug resistance, toxicity, high cost) [7]. Insecticide-based methods, i.e. indoor residual spraying (IRS), insecticide-treated bed nets, insecticide durable wall lining, etc, have achieved focal reduction of sand fly populations in highly endemic regions, yet primarily within the framework of mosquito vector control programs, rather than targeted regional sand

fly control campaigns [8]. Nevertheless, the selection pressure imposed by intensive insecticide applications for public health protection and/or off-target effects from agriculture results in the development of insecticide resistance (IR) in sand flies [9,10]. There is fragmented assessment of sand fly vectors resistance to insecticides and limited molecular data for the underlying resistance mechanisms. The majority of those resistance records are aggregated in leishmaniasis hyper-endemic regions of southeastern Asia and the Middle East, and refer mainly to DDT and pyrethroids [11], with the latter being the primary class of insecticides deployed in public health interventions [10]. Resistance phenotypes have been often associated with the presence of the voltage-gated sodium channel (*vgsc*) mutations L1014F/S (knockdown resistance mutations; *kdr*) [12,13].

At present, there are limited IR monitoring tools specialized for sand flies. World Health Organization (WHO) recently established the discriminating concentrations for WHO tube bioassays against currently used insecticides in five phlebotomine species (i.e. *P. papatasi*, *P. longipes*, *P. duboscqi*, *P. argentipes* and *L. longipalpis*) [14,15], while no specific CDC bottle bioassay guidelines are available [16]. A few studies so far have been conducted to assess the baseline susceptibility levels in CDC bioassays, often with contradictory results [17–19]; nevertheless, fundamental data for most sand fly vector species are missing. Furthermore, current bioassays for monitoring phenotypic resistance in wild vector populations (WHO tube, CDC bottle) are based on pooling samples of same vector species in single vials/tubes [14,15,20]. While for mosquitoes this is plausible since species discrimination relies on external morphology and wild-caught specimens can quickly and easily be grouped to genus and/or species, it may not readily apply to sand flies. The reason being that sand fly identification depends on internal morphology requiring dissection and mounting of specimens, thus current IR monitoring protocols need to be adjusted to allow for discriminating sand fly species when collections are made in regions with mixed species composition.

Contrary to mosquito vectors, where research has revealed a variety of IR mechanisms (i.e. target-site mutations, elevated detoxification, reduced cuticle penetration, behavioural avoidance and insecticide sequestration or excretion) [21–24], the paucity of genomic resources for sand flies seriously impedes deep investigation of such molecular basis, as well as the development of relevant diagnostic markers. All the above limitations critically hinder regular IR monitoring, hence, evidence-based management efforts in sand fly populations.

In order to enhance the scientific understanding of IR in sand fly populations in areas with limited data, we focus on the eastern Mediterranean basin and the Middle East, particularly on Greece, Turkey and Iran. All three countries share a long record of autochthonous leishmaniases transmission, while last decade's epidemiological data denote an escalating trend of visceral (VL) and cutaneous leishmaniasis (CL) prevalence, in Greece, and in Turkey and Iran, respectively [25–27]. Despite the rich sand fly diversity, certain vector species are more widely distributed and primarily implicated in parasite circulation in each country. *Phlebotomus perfiliewi*, *P. tobbi* and *P. neglectus* mediate *L. infantum* transmission in Greece [28], while *P. alexandri* is also dominant along with those aforementioned VL vectors in Turkey [29]. *Phlebotomus papatasi* and *P. sergenti* serve as the main vectors of CL causative agents, *L. major* and *L. tropica*, in Turkey and Iran [29–31].

Here, our main objectives are: i) monitoring sand fly species composition in populations collected from 12 regions of Greece, Turkey and Iran; ii) analysing pyrethroid resistance associated *Vgsc* gene loci in these populations; iii) phenotypically assessing by CDC bottle bioassays the response to deltamethrin of three mixed species sand fly populations from Greece; iv) designing a novel individual-specimen bioassay (a modified version of the CDC bottle bioassay), proposed to operate in regions with mixed sand fly species composition, and applying it here to compare three distinct *Phlebotomus* species against deltamethrin. Our molecular and

bioassay data enrich the scientific evidence for the IR status of distinct sand fly populations in areas of countries with overall poor data coverage, aiming to support evidence-based vector control efforts.

## Materials and methods

### 1. Sampling areas, sand fly collection and sample handling

Multiple sand fly collections were performed in regions of Greece, Turkey and Iran, between 2020 and 2023. Locations were selected in the interface between semi-rural environment, in agricultural areas and urbanized territories (e.g. residencies, streets), occasionally in close proximity to animal farms. Presence of stray or domestic dogs nearby, any known canine leishmaniasis (canL) case in the region and the history of insecticide applications (if known) were parameters taken into consideration, as well (Table 1 and Fig 1). Samples were collected from at least 2 to 3 different sites in each location, to avoid the probability of including isofemale sand flies in the downstream analyses.

In Greece, sand flies were collected overnight using CDC light traps baited with dry ice, set before 5 pm and removed at 7 am the day after (for two to ten consecutive night samplings). The collection bags were transferred from the field to the laboratory. In Iran, a hand aspirator and a head torch were used to collect sand flies resting on rocky and muddy surfaces at different time intervals between 8 pm and 5 am, and, afterwards, the samples were transferred from the field to the laboratory using cages. Both collection methods, i.e. CDC light traps and mouth aspirator, were combined in Turkish regions.

After field collections from Greece (in Attica, Thessaloniki, and Heraklion), we proceeded to CDC bioassays, excluding male and engorged female sand flies (that were only included in the following molecular analyses). Upon bioassay completion, females were stored in absolute ethanol for subsequent molecular analyses. For the remaining sampling locations in Greece (in Chania and Rethymno) and in all locations in Iran and Turkey, all the collected specimens were stored in ethanol for molecular analyses.

### 2 Genomic DNA extraction from sand flies

Genomic DNA (gDNA) was extracted from individual sand flies, using the DNazol reagent, according to the manufacturer's instructions (Invitrogen, Carlsbad, CA, USA). The DNA quantity was assessed by NanoDrop 2000c spectrophotometer.

### 3 Molecular identification of sand fly species

Discrimination of sand fly species relied on PCR amplification of a 700 bp mitochondrial cytochrome oxidase subunit I (*COI*) genomic fragment, a widely used molecular marker, using primers LCO1490 and HCO2198 [32] and *Taq* DNA polymerase (EnzyQuest, Heraklion, Greece) on approximately 10–20 ng of single specimen's gDNA template. The applied thermal protocol was as follows: 94°C for 2 min, 35 cycles x [94°C for 45 sec, 50°C for 30 sec, 72°C for 45 sec], 72°C for 10 min. After agarose gel visualisation of a small PCR product quantity, the rest was purified using the Nucleospin PCR & Gel Clean-Up Kit (Macherey Nagel, Dueren, Germany) and, then, subjected to Sanger sequencing (GENEWIZ, Azenta Life Sciences, Germany) with the LCO1490 primer and BLASTn analysis.

### 4 Genotyping of mutations in the *vgsc* gene

The presence of *kdr* mutations L1014F and L1014S, associated with resistance to pyrethroids and previously detected in sand flies and other insect species populations, was monitored in

**Table 1. Description of sand fly sampling locations in Greece, Turkey and Iran, their insecticide application history (if known), and the type of analyses (phenotypic and/or molecular) each collected population was subjected to.**

| Country | Province | Location (ID) | Coordinates | Site Description | Collection Date | Insecticide application history | Bioassays | Molecular analysis |
|---|---|---|---|---|---|---|---|---|
| TURKEY (TU) | Izmir | Eğridere (TU1) | (38.49895, 27.21685) | Semi-rural, inhabited area, collections within a sheepfold and a cow barn, surrounded by mudwalls | July 2022 | Occasional spraying of deltamethrin and bendiocarb against houseflies & mosquitoes in the cattle farm & the sheepfold | N/A | ✓ |
| | Adana | Sarıçam (TU2) | (37.04129, 35.39404) | Semi-rural area, collections within a sheepfold, surrounded by brick walls | June 2022 | Long history of insecticide usage (DDT & OPs), PYs applied as of 1990 against mosquito vectors | N/A | ✓ |
| | | Koyunevi (TU3) | (37.28691, 35.64977) | Semi-rural area, collections within a cow barn and a chicken coop, surrounded by mudwalls | June 2022 | Long history of insecticide usage (DDT & OPs), PYs applied as of 1990 against mosquito vectors | N/A | ✓ |
| | Gaziantep | Akçaburç (TU4) | (37.24701, 37.31289) | Rural area, collection within a cow barn surrounded by brick walls; olive groves, pistachio trees and chicken coops nearby | June 2022 | Intense PY usage for agricultural and public health (mosquito, housefly) purposes | N/A | ✓ |
| | Hatay | Hıdırbey (TU5) | (36.12799, 35.97163) | Rural area, collection within a cow barn surrounded by brick walls; olive groves and chicken coops nearby | June 2022 | Rare spraying of PY (mostly, α-cypermethrin & deltamethrin) | N/A | ✓ |
| IRAN (IR) | Lorestan | Sarab Hamman (IR1) | (33.1091667, 47.6930556) | Semi-rural area, cultivated land; collections near multi-species animal farms and agricultural fields | August 2022 | Insecticide spraying for agricultural purposes by farmers | N/A | ✓ |
| | Isfahan | Matin Abad (IR2) | (33.764596, 51.981421) | Rural, inhabited area, cultivated land; collections near rodent hosts | August 2022 | ITNs in a few places around, occasional baiting against rodent hosts | N/A | ✓ |
| | Kerman | Orzuiyeh (IR3) | (28.4413889, 56.3727778) | Semi-rural, inhabited area, cultivated land; collections near multi-species animal farms | August 2022 | Unknown | N/A | ✓ |
| GREECE (GR) | Attica | Kalentzi (GR1) | (38.173352, 23.917696) | Semi-rural, inhabited area, collection in abandoned & demolished house; olive groves and chicken coop in close proximity | September 2020 | Regional mosquito larviciding programs in Western Attica; possible additional use of insecticides against agricultural pests and/or at household level | ✓ | ✓ |
| | Thessaloniki | Thermi, American Farm School (GR2) | (40.571823, 22.986729) | Cattle and poultry farms, crop fields | September 2020 & September 2022 | PYs & IGRs sprayed bi-monthly around the farms | ✓ | ✓ |
| | | Thermi, Organic farm (GR3) | (40.5578265, 23.0262722) | Semi-rural area, collections within multi-species animal farm (sheep, poultry, horse etc); stray dogs and olive groves around | September 2020 & September 2022 | No use of insecticides | ✓ | ✓ |
| | Heraklion | Agia Pelagia (GR4) | (35.3950446, 24.9906789) | Semi-rural, hilly area, collection within inhabited villa's yard; domesticated dogs, chicken coop & olive groves around | September 2020 | Unknown; possible use of insecticides for regional agricultural activities and/or at household level | ✓ | ✓ |
| | Rethymno | Mylopotamos (GR5) | (35.405725, 24.696072) | Rural area, cultivated land, collection in olive grove; inhabited villas in close proximity | September 2023 | Unknown; possible use of insecticides for regional agricultural activities and/or at household level | N/A | ✓ |
| | Chania | Vamvakopoulo (GR6) | (35.494246, 23.986598) | Semi-urban area, agricultural activities around; collection in olive grove | September 2023 | Unknown; possible use of insecticides for regional agricultural activities and/or at household level | N/A | ✓ |

N/A, not applicable; PYs, pyrethroids; OPs, organophosphates; IGRs, insect growth regulators; ITNs, insecticide-treated nets. Information on the insecticide application history of each location was obtained from local farmers and shepherds.

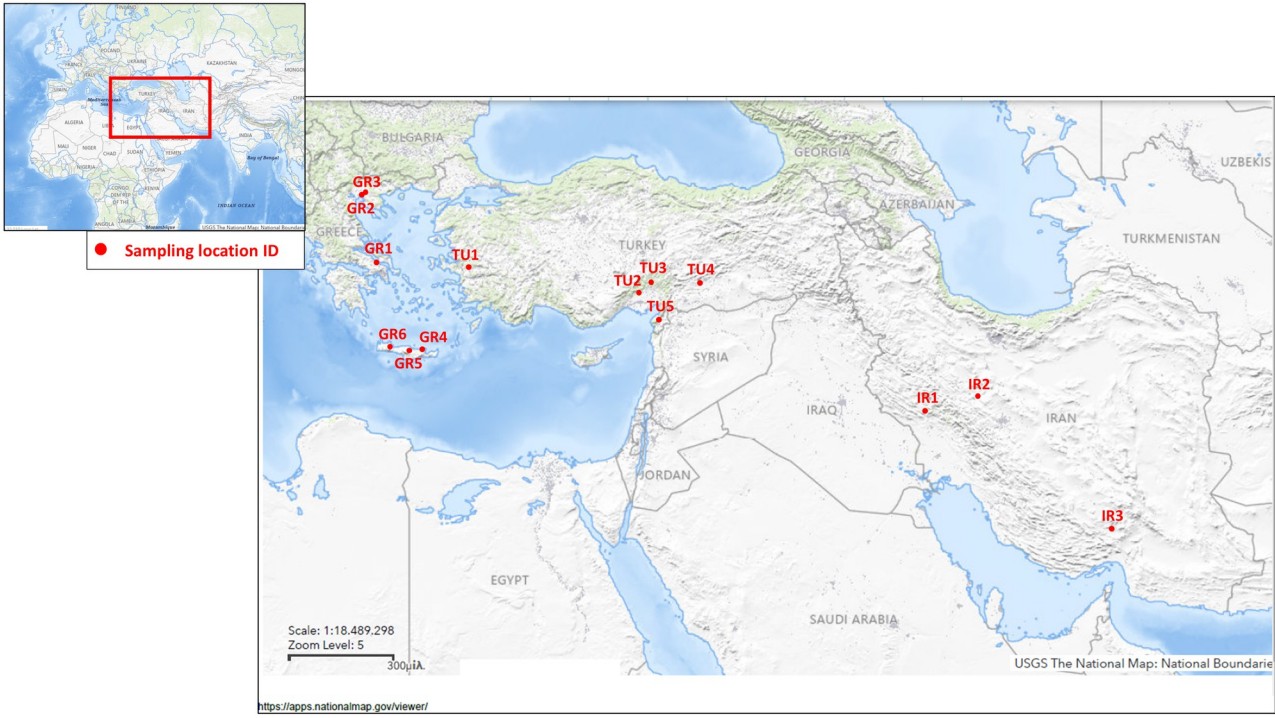

**Fig 1. Sand fly sampling locations in Greece (GR1-GR4), Turkey (TU1-TU5) and Iran (IR1-IR3).** Maps were obtained from U.S. Geological Survey (https://apps.nationalmap.gov/viewer/).

individual sand flies by genotyping the *vgsc* domain IIS6, containing codon 1014 (*Musca domestica* numbering). The genomic sequence was amplified by *Taq* DNA polymerase (Enzy-Quest, Heraklion, Greece) using primers Vssc8F and Vssc1bR, as described in Gomes et al., 2017 [12], on approximately 10–20 ng of gDNA template. The reaction's thermal conditions were: 94˚C for 2 min, 35 cycles x [94˚C for 45 sec, 56˚C for 30 sec, 72˚C for 30 sec], 72˚C for 10 min. The approximately 400 bp generated PCR fragments, after visualization in agarose gel, were purified using the Nucleospin PCR & Gel Clean-Up Kit (Macherey Nagel, Dueren, Germany) and, then, subjected to Sanger sequencing (GENEWIZ, Azenta Life Sciences, Germany), using the Vssc8F primer. Sequences were analysed using the sequence alignment editor BioEdit 7.2.5 (https://bioedit.software.informer.com/7.2/). Reference *vgsc* partial genomic sequences were obtained from GenBank for *Phlebotomus papatasi* (MH401419.1), *P. perfiliewi* (MG779187.1), *P. neglectus* (MG779192.1), *P. simici* (MG779189.1) and *P. tobbi* (MG779188.1).

## 5 Insecticide susceptibility bioassays

**i Standard CDC bottle bioassay.** Female, non-engorged field-caught sand flies from Kalentzi (GR1, Attica), Thermi (GR2, GR3, Thessaloniki) and Agia Pelagia (GR4, Heraklion), Greece, were exposed to deltamethrin (pyrethroid insecticide), purchased as technical grade material (PESTANAL analytical standard; Sigma-Aldrich, Darmstadt, Germany). Insecticide stock solutions were prepared in acetone and 250 ml Wheaton bottles were coated as described in the CDC guidelines [20, 33]. The insecticide doses tested for the Greek populations were the CDC diagnostic for mosquito vectors; particularly, deltamethrin 10 μg/ml (diagnostic dose for

*Aedes* mosquitoes before 2021; applied to all four populations, in 2020 [20]) and/or 0.75 μg/ml (current diagnostic dose for *Aedes* and *Culex*; applied to populations GR2 and GR3, from Thermi, in 2022). Two to six replicate bottles, including 15 to 20 individuals each, were tested for any given deltamethrin concentration (the number of replicates per population was relative to the number of collected sand flies) and a control bottle coated with acetone was always included. Alive and knocked-down sand flies in each bottle were recorded at time intervals of 5 minutes, for 1 hour exposure time. The percentage of mortality in each bottle was recorded after 24 hours of recovery; during that period, sand flies, transferred in plastic cups, were maintained in temperature and relative humidity (RH) conditions, as close to the optimal as possible (~25˚C, RH 70%), supplemented with 10% sucrose solution. The insecticide susceptibility status was determined by the mortality rate, according to CDC recommendations: 98–100% mortality indicates susceptibility; 90–97% suggests the possibility of resistance that requires further confirmation; and mortality < 90% denotes resistance. In cases where mortality (between 3–5%) was recorded in the control bottle, mortality data were corrected using Abbott's formula. All specimens were stored in ethanol for subsequent molecular analyses, as described above.

**ii Individual specimen glass vial bioassay (modified CDC bottle bioassay).** The objective behind generating this novel individual specimen bioassay protocol for sand flies is to detect any possible hint of tolerance/resistance to insecticides in single species, that might have been "diluted" during mixed species exposure. The protocol generated is a modified version of the CDC bottle bioassay protocol, adjusted in 9 ml round bottom, plain-end glass tubes of dimensions 12 x 100 mm (Fisherbrand). 24-hour viability of female sand flies inside these vials was tested using *Phlebotomus papatasi* laboratory colony (EBCL, USDA) specimens.

To apply insecticide doses equivalent to the CDC diagnostic ones (i.e. 1 ml of deltamethrin solution 0.75 μg/ml to coat a 250 ml Wheaton bottle), we needed 36 μl of 0.75 μg/ml deltamethrin solution for 9 ml glass vials' coating, a volume which, during initial coating trials, was proved to be insufficient to fully coat the inner walls of them. Thus, we doubled the volume to 72 μl and prepared a 0.375 μg/ml deltamethrin solution. Similar modifications of the standard CDC bioassay protocol were previously described in Denlinger et al. 2015 [17] and Li et al. 2015 [18].

Control vials were coated with 72 μl of acetone and all vials were left to dry overnight in a shady place. The next day, using a mouth aspirator, individual females were transferred inside insecticide-treated and control vials, sealed with cotton (S1 Fig). Each specimen's survival was observed every 5 minutes and the knock-down time (KDT) was recorded. The standard exposure to deltamethrin was 1 hour of exposure, and, then, the specimens were transferred individually in carton cups, supplemented with 10% sucrose solution. Individuals showing more tolerance to deltamethrin (KDT over 60 min) were exposed for 2 hours totally. Mortality was recorded after 24 hours of recovery in standard insectary conditions, and all specimens were separately stored in absolute ethanol for subsequent molecular identification of species and genotyping of resistance mutations, as described above.

## Results

### 1 Molecular identification of sand fly species

Species discrimination of the collected sand flies was *COI*-based and determined from over 98% identity to the genomic sequences deposited into GenBank.

Six different *Phlebotomus* and two *Sergentomyia* species were recorded in the five regions of Greece (Table 2). In Kalentzi, Attica, *P. neglectus* and *P. simici* were the most prevalent species (37.3% and 21.6%, respectively), followed by *P. tobbi*. *Phlebotomus perfiliewi* was exclusively

**Table 2. Sand fly species composition (%) per sampling region.**

| | Region | N | P. papatasi | P. neglectus | P. perfiliewi | P. tobbi | P. simici | P. similis | P. sergenti | P. major | P. jacusieli | S. minuta | S. dentata |
|---|---|---|---|---|---|---|---|---|---|---|---|---|---|
| | | | | | | | % Species composition (n) | | | | | | |
| GREECE | Attica | 51 | - | 37.3 (19) | - | 13.7 (7) | 21.6 (11) | - | - | - | - | 11.8 (6) | 15.7 (8) |
| | Thessaloniki | 211 | 0.5 (1) | 1.0 (2) | 44.5 (94) | 27.9 (59) | 23.6 (50) | - | - | - | - | 1.9 (4) | 1.0 (2) |
| | Heraklion | 53 | - | 35.8 (19) | - | - | 50.9 (27) | - | - | - | - | 13.2 (7) | - |
| | Rethymno | 45 | 4.4 (2) | 51.1 (23) | - | - | 4.4 (2) | 35.6 (16) | - | - | - | 4.4 (2) | - |
| | Chania | 8 | - | 25.0 (2) | - | - | - | 37.5 (3) | - | - | - | 37.5 (3) | - |
| TURKEY | Izmir | 5 | 80 (4) | - | - | 20 (1) | - | - | - | - | - | - | - |
| | Adana | 23 | 78.3 (18) | - | - | 4.3 (1) | 4.3 (1) | - | - | 4.3 (1) | - | - | 8.7 (2) |
| | Gaziantep | 26 | 38.5 (10) | - | 19.2 (5) | 15.4 (4) | 7.7 (2) | - | 15.4 (4) | 3.8 (1) | - | - | - |
| | Hatay | 12 | 16.7 (2) | - | - | 41.7 (5) | - | - | - | 33.3 (4) | 8.3 (1) | - | - |
| IRAN | Matin Abad | 72 | 100 (72) | - | - | - | - | - | - | - | - | - | - |
| | Sarab Hamman | 22 | 100 (22) | - | - | - | - | - | - | - | - | - | - |
| | Orzuiyeh | 50 | 100 (50) | - | - | - | - | - | - | - | - | - | - |

N corresponds to the total number of sand flies identified molecularly to species per population and n given in brackets refers to the absolute number of samples per species in each population. Species composition in the regions of Thessaloniki and Adana is presented cumulatively for the sampling sites American Farm School and Organic Farm, and Sarıçam and Koyunevi, respectively

detected in Thermi, Thessaloniki, where it appeared dominant with a frequency over 40%. In the same region, more than half of the samples belonged to *P. tobbi* or *P. simici* species. In the Prefecture of Crete (Heraklion, Rethymno and Chania regions), *P. neglectus*, *P. simici* and *P. similis* denoted the highest occurrence among *Phlebotomus* species. *P. papatasi* was detected only in Thessaloniki and Rethymno, in frequencies below 5%. Lastly, *S. minuta* was present in all sampling regions of the country, peaking at Chania (37.5%), and *S. dentata* only in Attica and Thessaloniki, holding frequencies of 15.4% and 1.0% respectively.

*Phlebotomus* genus in Turkish populations was principally represented by *P. papatasi* (38.5–80%) in three out of four sampling locations. *Phlebotomus tobbi* frequencies ranged from 4.3 to 20% in most locations, except Hatay where they reached ~ 40%. Other *Phlebotomus* species, particularly *P. perfiliewi*, *P. simici*, *P. sergenti* and *P. jacusieli*, were also present in some locations with frequencies below 20%. *P. major* was reported in all regions apart from Izmir (overall very low number of collected specimens in this region), reaching a maximum frequency of almost 35% in Hatay. *S. dentata* was the sole *Sergentomyia* species detected in Turkey (<10% in Adana) (Table 2).

In Iran, all collected samples were exclusively identified as *P. papatasi*.

## 2. Monitoring of pyrethroid resistance mutations *kdr* L1014F/S

*Phlebotomus* and *Sergentomyia* female and male samples from all regions were analyzed for the presence of *vgsc* mutations L1014F and L1014S, associated with pyrethroid resistance (Table 3).

Mutant allele 1014F (codon TTT) held a surprisingly high frequency, reaching almost 80% in Adana, Turkey, while occurring in homozygosity in 72.2% of the samples; nearly all 1014F/1014F homozygotes belonged to *P. papatasi*, except for one *S. dentata*. A single *P. tobbi* specimen harboured 1014F in heterozygosity (1014L/1014F; TTA/TTT). Mutation 1014S (serine encoded by TCA) was additionally recorded in Adana (frequency <3%), in a heterozygote (1014F/1014S; TTT/TCT) *P. papatasi* individual. Moreover, in Gaziantep, 1014F was the only

**Table 3. Molecular monitoring of *kdr* mutations L1014F/S in the collected sand fly populations.**

| Country | Region | N | Allele L / F / S (%) | Genotype (%) |
|---|---|---|---|---|
| GREECE | Attica | 27 | L 100 | LL 100 |
| | Thessaloniki | 156 | L 100 | LL 100 |
| | Heraklion | 35 | L 100 | LL 100 |
| | Rethymno | 42 | L 97.6 S 2.4 | LL 97.6 SS 2.4 |
| | Chania | 8 | L 100 | LL 100 |
| TURKEY | Izmir | 4 | L 100 | LL 100 |
| | Adana | 18 | L 19.4 F 77.8 S 2.8 | LL 16.7 LF 5.6 FF 72.2 FS 5.6 |
| | Gaziantep | 22 | L 84.1 F 15.9 | LL 77.3 LF 13.6 FF 9.1 |
| | Hatay | 11 | L 100 | LL 100 |
| IRAN | Matin Abad | 72 | L 100 | LL 100 |
| | Sarab Hamman | 22 | L 88.6 F 11.4 | LL 77.3 LF 22.7 |
| | Orzuiyeh | 50 | L 100 | LL 100 |

N corresponds to the total number of sand flies genotyped individually for L1014F/S mutations. Genotyping data for Thessaloniki and Adana are presented cumulatively for the sampling sites American Farm School and Organic Farm, and Sarıçam and Koyunevi, respectively. Alleles: L, 1014L, wild-type; F, 1014F, mutant; S, 1014S, mutant. Genotypes: LL, homozygote 1014L/1014L; LF, heterozygote 1014L/1014F; FF, homozygote 1014F/1014F; FS, heterozygote 1014F/1014S; SS, homozygote 1014S/1014S.

mutant allele present, in a frequency of 15.9%, with two *P. papatasi* homozygotes (TTT/TTT) and three 1014L/1014F (TTA/TTT) heterozygote samples; two *P. papatasi* and one *P. perfiliewi*. All samples from Izmir and Hatay were wild-type (1014L/1014L; TTA/TTA).

In Sarab Hamman, the only *kdr*-positive population from Iran, 5 out of the 22 genotyped *P. papatasi* individuals carried 1014F allele in heterozygosity (1014L/1014F encoded by TTA/TTT; mutant allele frequency 11.4%).

In Greece, no population harboured the mutation L1014F, while only one *P. papatasi* specimen from Rethymno (GR5) was homozygote for 1014S (TCT/TCT, frequency 2.4%).

## 3. Insecticide susceptibility bioassays

### i. Standard CDC bottle bioassays

Mixed species populations from Greece, i.e. Kalentzi (GR1, Attica), Thermi (GR2 and GR3, Thessaloniki) and Agia Pelagia (GR4, Heraklion), collected in 2020, were exposed to deltamethrin 10 μg/ml for 1 hour. Fifty percent of all populations were knocked down within the first 20 minutes of exposure, and no significantly different knock-down rate was noted among them (Fig 2A). Mortality recorded after 24 hours was 100% for all four populations, indicating susceptibility, according to CDC guidelines (Fig 2B).

During collection year 2022, to exclude the possibility of any upcoming resistance/ tolerance in the regularly pyrethroid-treated American Farm School population, Thermi, (GR2), we exposed this population comparatively to its proximal Organic Farm population (GR3), to a much lower deltamethrin dose, 0.75 μg/ml, for 1 hour. Both populations displayed an almost

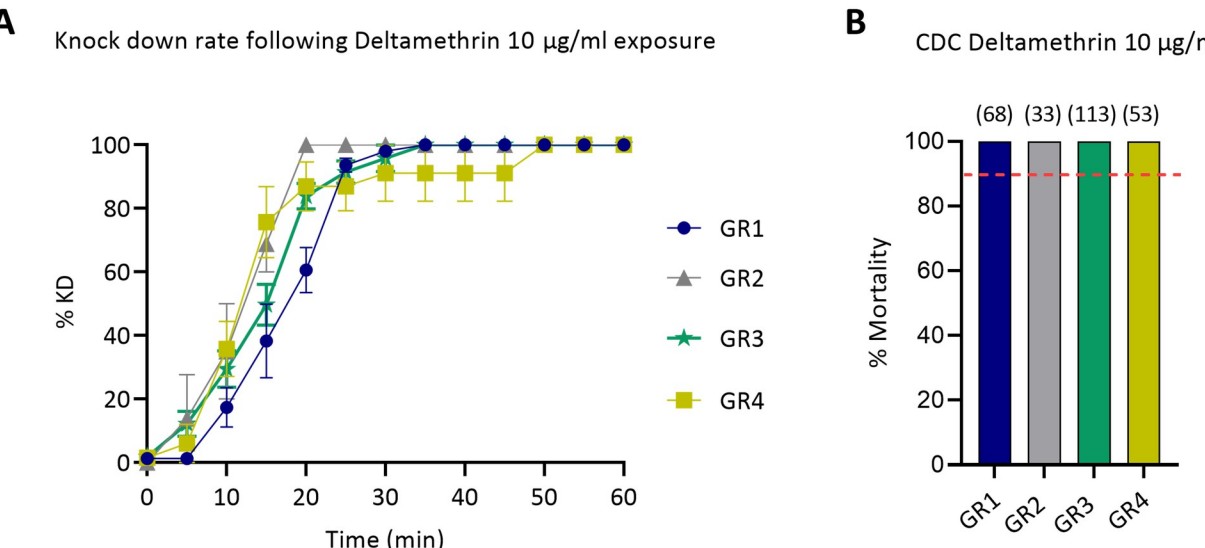

**Fig 2. Response of GR1-GR4 mixed species populations from Greece, to deltamethrin 10 μg/ml in CDC bottle bioassays.** A) Knock-down rate recorded during one hour exposure to deltamethrin. Results are expressed as the mean knock down percentage (%) and bars represent SEM (n = 2 to 5 biological replicates). B) Percentage (%) of mortality recorded 24 hours post exposure. Results are expressed as the mean mortality percentage (%) and the total number of exposed specimens per population are given in brackets. The red dotted line corresponds to the 90% mortality cut off indicating resistance, according to CDC guidelines.

identical response and, as expected, a slower knock down rate compared to the 10 μg/ml exposure (Fig 3A), yet 100% mortality (Fig 3B).

### ii. Individual specimen glass vial bioassays

Specimens collected, during summer 2022, from Thermi, Thessaloniki, were exposed individually to deltamethrin. American Farm School and Organic Farm populations were considered here as a single population, given their close proximity, their similar species composition and the absence of any significant difference in their knock-down rate, in CDC deltamethrin 0.75 and 10 μg/ml bioassays (see Figs 2 and 3). The insecticide dose was 0.75 μg/ml and was chosen based on the previous observations during CDC bioassays, where sand flies from the same locations exhibited a "smoother" knock-down rate, compared to when exposed to 10 μg/ml, making any differential response more noticeable.

During deltamethrin treatment, the earliest knock-down response started from 9 min exposure, while the longest KDT noted was 120 min; approximately 83% of the samples were knocked down within the first hour of exposure. All samples were dead after 24 hours of recovery. Following molecular identification of species, it was revealed that the mean KDT of *P. simici* samples (N = 34) was approximately 60 min, significantly higher from *P. perfiliewi* KDT 35.3 min (**, p = 0.001; N = 89). *P. tobbi* denoted an intermediate mean KDT of 43.3 min (N = 32), not significantly different from both *P. perfiliewi* and *P. simici* (Fig 4).

### Discussion

Leishmaniasis mitigation efforts, largely relying on sand fly control by means of insecticide-based tools, are threatened by the development of IR phenotypes in vectors' populations [34]. This study represents an attempt to address this critical knowledge gap in sand flies by developing and applying entomological and IR profiling tools, in populations from leishmaniasis

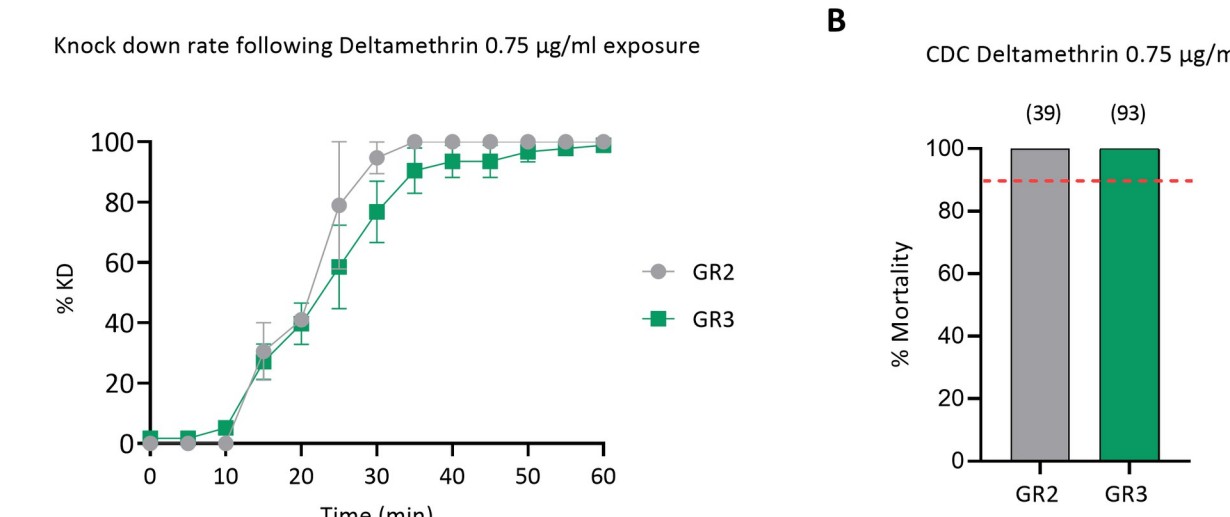

**Fig 3. Response of GR2 and GR3 mixed species populations from Thermi, Thessaloniki, to deltamethrin 0.75 µg/ml in CDC bottle bioassays.** A) Knock-down rate recorded during one hour exposure to deltamethrin. Results are expressed as the mean knock down percentage (%) and bars represent SEM ($n$ = 2 to 5 biological replicates). B) Percentage (%) of mortality recorded 24 hours post exposure. Results are expressed as the mean mortality percentage (%) and the total number of exposed specimens per population are given in brackets. The red dotted line corresponds to the 90% mortality cut off indicating resistance, according to CDC guidelines.

endemic countries of the Mediterranean basin and the Middle East with limited monitoring data, i.e. Greece, Turkey and Iran. Our sampling locations were in (semi-)rural regions, with agricultural activities and/or livestock farming, in the vast majority of which insecticides are applied mainly for crop protection and/or occasionally for public health purposes (e.g. targeting mosquito vectors), at regional and/or at farm/household level.

Molecular identification of species in the collected populations appeared largely in line with previously reported data [29,35–39]. *Phlebotomus neglectus* was among the most frequently detected species in Attica and in the Island of Crete (Heraklion, Rethymno and Chania) [35,36], Greece, while *P. perfiliewi* was exclusively and predominantly recorded in Thessaloniki [37]. These two species, along with *P. tobbi* (found here in Attica and Thessaloniki in frequencies lower than 30%), serve as main *L. infantum* vectors in the respective regions of the country; VL is mostly of veterinary concern, as canL prevalence is predicted to exceed 50% [25, 28]. Moreover, and in accordance with data generated by Chaskopoulou et al., 2016 [37] and Dvorak et al., 2020 [36], *P. similis* appeared exclusively in Crete, reaching almost 40% in Rethymno, and *P. simici* was relatively abundant across the whole country; both species consist yet unproven vectors of CL and VL, respectively. *Sergentomyia minuta* was present in all Greek locations, as expected, in moderate to low frequencies, whereas *S. dentata* only in the mainland of the country.

In Turkey, *P. papatasi*, primarily transmitting CL, was, as expected, the most prevalent species in three out of four surveyed Provinces, with Adana and Gaziantep listed among those reporting the highest number of human cases [38]. Conforming with Kasap et al 2019 [29], *P. tobbi* comes second in frequency and distribution, serving as the main *L. infantum* vector there, denoting the highest occurrence in Hatay and Izmir, in the Mediterranean part of Turkey that carries the heaviest canine and human VL load [38]. Various other species, i.e. *P. perfiliewi*, *P. sergenti*, *P. simici*, *P. jacusieli* and *S. dentata*, were detected in frequencies below 20%, in the Mediterranean/ south-eastern sampling locations of the country. In the sampling

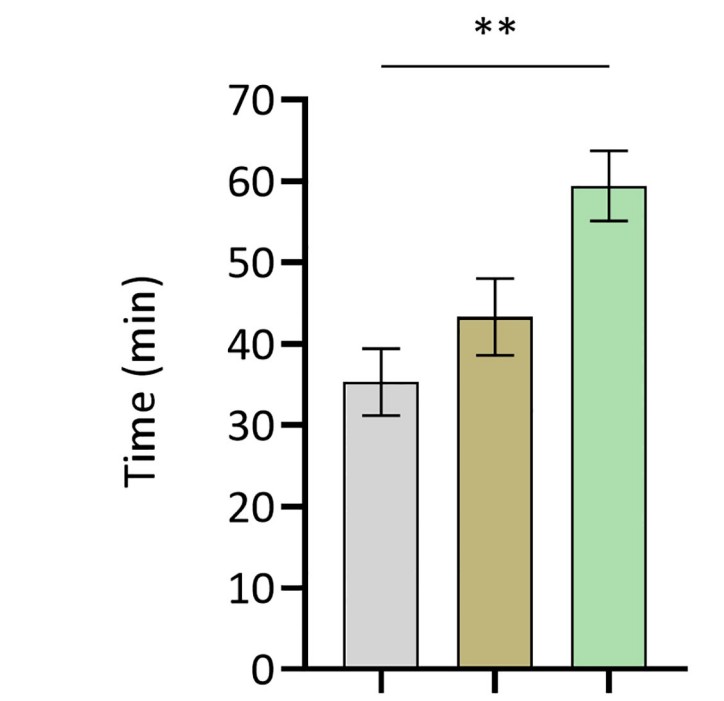

**Fig 4. Mean knock down time (KDT) of three *Phlebotomus* species populations, i.e. *perfi*, *P. perfiliewi*; *tob*, *P. tobbi*; *simc*, *P. simici*, from Thermi, Greece, upon exposure to deltamethrin 0.75 μg/ml, in individual glass vial bioassays.** *N* corresponds to the total of individually exposed female specimens per species, cumulatively from American Farm School and Organic Farm. Bars represent SEM. Results were statistically analysed using one-way ANOVA; **, $p = 0.001$.

locations from Iran, *P. papatasi* was the sole species identified being the main zoonotic CL vector, widely distributed across the whole country [31, 39] Notably, *P. sergenti*, mediating anthroponotic CL transmission in Iran, was not detected here, even though Isfahan and Kerman are among the provinces with the highest annual CL incidence (48.8 and 25.3 per 100,000 population, respectively, on average between 2013–2020) [27]; species' microhabitat preferences could possibly account for this result [40].

It needs to be clarified though that our species identification data neither reflect the integrated species composition of each region nor the seasonal abundance of species, since we focused on specific environments/microhabitats (as described in Table 1), where we conducted a few consecutive night samplings, rather than systematic entomological surveillance.

Besides that, different collection methods (i.e. CDC light traps / mouth-aspirator), might have also impacted the species structure and/or abundance. Nonetheless, it is encouraging to observe that the populations' composition resulting from these targeted and relatively limited collection interventions is in agreement with data reported from previously conducted, more intensive surveillance efforts.

Interestingly, pyrethroid resistance mutations were recorded in all three surveyed countries, albeit in different frequencies. In Adana and Gaziantep, regions with long history of pyrethroid usage in Turkey, L1014F was recorded mainly in *P. papatasi*, particularly prevalent in the former region (frequency >70%). This mutation was previously identified in same species samples from Sanliurfa [41], an area close to our sampling regions in the south-eastern part of the country. In addition to that, mutation L1014S was noted for the first time in the country, in one 1014S/1014F *P. papatasi* sample from Adana, while in Izmir and Hatay regions, no mutation was recorded, yet the sample sizes were very small. *Kdr* L1014S and L1014F occurred likewise in *P. papatasi* specimens from Rethymno, Greece, and Sarab Hamman, Iran, respectively, albeit at frequencies below 12%. To the best of our knowledge, this is the first study to detect *kdr* mutations in sand fly populations from Greece and Iran, although certain regions of each country have been surveyed before [41–43].

The occurrence of resistance mutations, in our study sites, has probably followed pyrethroid and/or DDT usage (as *kdr* mutations confer cross-resistance to both) implemented in the respective rural locations (e.g. regions Adana, Gaziantep, Sarab Hamman, etc), for agricultural protection and/or mosquito control. Molecular markers could have an important predictive value and ease the early detection of developing resistance [44]. As of 2017, several studies have monitored *kdr* mutations in sand fly populations from eight leishmaniasis endemic countries, in the Mediterranean basin, the Middle East and south-eastern Asia [11, 45–47]. In all of the surveyed countries, apart from Italy [48], i.e. India, Sri Lanka, Bangladesh, Iran, Armenia, Turkey and Greece, L1014F and/or L1014S were indeed present. Such findings highlight that systematic regional IR surveillance and management are necessary, as pyrethroid resistance might be spreading in sand fly populations, as is the case for malaria vectors [10]. IRS predominantly with DDT, and pyrethroids, used to be the mainstay of leishmaniasis control globally, whereas carbamates and organophosphates partially/ occasionally replace this insecticide application regime lately [10, 49].

Besides molecular diagnostics, CDC bottle bioassays were deployed to assess the susceptibility to deltamethrin of mixed sand fly species populations collected from Kalentzi, Thermi, and Agia Pelagia, Greece. Extrapolating the diagnostic doses of mosquitoes (10 μg/ml and, later, 0.75 μg/ml), they displayed a "susceptible" phenotype, with no notable difference in their knock-down rate during exposure, concurring with the absence of associated *kdr* mutations. Even at the lowest tested dose, the American Farm School and the nearby Organic Farm populations, sharing the same species structure, yet polar opposite insecticide exposure records, responded in a same manner, possibly implying that no apparent resistance to deltamethrin has been developed over the years in the regularly pyrethroid-sprayed AFS population. Unfortunately, the small sample sizes from Iran and Turkey impeded conducting bioassays and potentially linking the presence of *kdr* alleles (especially where recorded in high frequencies, i.e. Turkey) to pyrethroid resistance phenotype. Such information would be valuable to mirror any possible impact on the phenotype, since sole occurrence of *kdr* mutations does not necessarily confer pyrethroid survival in wild populations [50,51]. Nonetheless, we need to consider, that sand fly bioassay results using mosquito discriminating doses might be difficult to interpret [11,16]; comparing the current WHO tube bioassay discriminating doses recommended for mosquitoes to those for sand flies [14,15], vector-specific differences are easily perceived in several insecticide compounds therefore the operational value of the generated data might be

limited for targeted vector control. Any upcoming tolerance/ resistance is likely to pass unnoticed or be over-estimated in case the tested dose is too high or too low, respectively, for sand fly vectors *per se*.

Another critical point in phenotypic monitoring of IR is that exposing a mixed species population in a single CDC bottle (or WHO tube) might conceal possible divergent phenotypes of single species [52]. Proceeding to individual oviposition of field-caught females to create laboratory-reared isofemale progenies and, then, to bioassays, entails certain difficulties and risks, as discussed by Shirani-Bidabadi et al., 2020 [19]. Herein, to identify any obscured signs of tolerance/resistance to insecticides in single species, an alternative bioassay protocol to expose single sand fly specimens in insecticide-coated glass vials right after field collections is proposed. Applying this protocol to the samples collected from Thermi, Greece, revealed that *Phlebotomus* species might indeed hold differing responses to insecticides. Particularly, *P. simici* exhibited an almost 2x-fold higher KDT than *P. perfiliewi* upon treatment with 0.75 μg/ ml deltamethrin (mean KDT 59.4 vs 35.3 min). *Phlebotomus simici*, a suspected *L. infantum* vector and among the most dominant species in our collections from the mainland of Greece, is well-established in the eastern Mediterranean basin and was recently recorded for the first time in Austria, raising concerns on its possible northward geographical expansion [53]. Although we perceive that this is a time-consuming and technically difficult methodology to be incorporated into routine vector control campaigns, it can contribute valuable information for efficient species-specific control in relevant eco-epidemiological settings, even if applied in a sub-group of samples, complementary to regular CDC bottle bioassays.

Nevertheless, it is essential to point out that resistance traits can be focally distributed, following the insecticide selection pressure regimes in each location, and thus probably rare to detect, especially given the mixed species composition [54], and the confined spatial movements that certain sand fly species might display [55]. Hence, we cannot rule out the possibility that target-site resistance traits (or detoxification-based, that were not examined here) may similarly occur in the rest of the analysed collections, especially those represented by a small sample size (such as Rethymno and Chania, Greece, and Izmir and Hatay, Turkey).

Monitoring the development of IR in sand flies is a core-element of integrated vector management (IVM), providing evidence-based guidance to vector control campaigns in an eco-epidemiologically context-specific manner. Following the paradigm of mosquito surveillance, for which the existing IR profiling/ reporting databases (e.g. IR Mapper) have greatly supported decision making, developing similar tools for sand flies and tailoring *Leishmania* infection surveillance systems to IVM would benefit sustainable leishmaniasis control activities [34].

## Conclusion

In short, this study enriches the IR status map of sand fly populations in regions of the Mediterranean basin and the Middle East, poorly monitored thus far. Molecular analyses of pyrethroid resistance in populations collected from Greece, Turkey and Iran, focused on *kdr* mutations, lately often reported in leishmaniasis endemic regions. The occurrence of L1014F/ S mutations is more prevalent in Turkey and in lower frequencies, yet recorded for the first time, in Greece and Iran, implying emerging resistance. Phenotypic assessment of some mixed species populations from Greece against deltamethrin indicated susceptibility, extrapolating though the diagnostic doses for mosquitoes, an approach that usually leads to difficult-to-evaluate/-interpret results. Added to these, by establishing a novel bioassay protocol for individual specimens, we managed to show that sand fly species could display significantly different response to insecticides, highlighting the importance of resistance monitoring and

regional vector control to be context- and species- specific. Given the limited arsenal of insecticide compounds appropriate for public health protection, sand fly control should be evidence-based, relying on regular species composition and IR monitoring, the latter encompassing both molecular and bioassay data, derived from sand fly-standardized protocols. Last but not least, there is an urgent need to generate relevant genomic resources for sand flies, and develop high-throughput methodologies to characterise IR mechanisms, that would also be beneficial for the identification of novel resistance diagnostic markers.

## Supporting information

**S1 Fig. Sand fly exposure to deltamethrin, in individual glass vials bioassays, compared to standard CDC bottle bioassays.** Authors' personal photograph collection during experiments. Image " a " by Sofia Balaska, September 2022; Image " b " by Latifa Remadi, September 2022. (DOCX)

**S1 Data.** A. *COI*-based species identification and genotyping of kdr mutations in samples collected from Iran; B. *COI*-based species identification and genotyping of kdr mutations in samples collected from Turkey; C. *COI*-based species identification and genotyping of *kdr* mutations in samples collected from Greece; D. Standard CDC bottle bioassays against deltamethrin 10 µg/ml in mixed species populations collected from Greece: knock-down rate and % mortality; E. Standard CDC bottle bioassays against deltamethrin 0.75 µg/ml in mixed species populations collected from Greece: knock-down rate and % mortality; F. Individual exposure of sand flies from Thermi, Thessaloniki, to deltamethrin 0.75 µg/ml, in glass vials: knock down time, and subsequent identification of species based on *COI*. (XLSX)

## Author Contributions

**Conceptualization:** Sofia Balaska, Konstantinos Mavridis, Alexandra Chaskopoulou, John Vontas.

**Data curation:** Sofia Balaska, Jahangir Khajehali, Konstantinos Mavridis, Mustafa Akiner, Kyriaki Maria Papapostolou, Latifa Remadi.

**Formal analysis:** Sofia Balaska.

**Funding acquisition:** Sofia Balaska, Konstantinos Mavridis, Latifa Remadi.

**Investigation:** Sofia Balaska, Konstantinos Mavridis, Mustafa Akiner.

**Methodology:** Sofia Balaska, Jahangir Khajehali, Konstantinos Mavridis, Mustafa Akiner, Kyriaki Maria Papapostolou, Latifa Remadi, Ilias Kioulos, Michail Miaoulis, Emmanouil Alexandros Fotakis.

**Project administration:** John Vontas.

**Resources:** Sofia Balaska, Jahangir Khajehali, Konstantinos Mavridis, Mustafa Akiner, Ilias Kioulos, Michail Miaoulis, Emmanouil Alexandros Fotakis.

**Supervision:** Alexandra Chaskopoulou, John Vontas.

**Validation:** Sofia Balaska.

**Visualization:** Sofia Balaska.

**Writing – original draft:** Sofia Balaska, John Vontas.

**Writing – review & editing:** Sofia Balaska, Konstantinos Mavridis, Alexandra Chaskopoulou, John Vontas.

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
