## [Decision Letter · Decision Letter 0]

3 Sep 2024

Dear Dr. Vontas,

Thank you very much for submitting your manuscript "Development and application of species ID and insecticide resistance assays, for monitoring sand fly Leishmania vectors in the Mediterranean basin and in the Middle East" for consideration at PLOS Neglected Tropical Diseases. As with all papers reviewed by the journal, your manuscript was reviewed by members of the editorial board and by several independent reviewers. The reviewers appreciated the attention to an important topic. Based on the reviews, we are likely to accept this manuscript for publication, providing that you modify the manuscript according to the review recommendations. 

Sincerely,

Yara M. Traub-Csekö

Academic Editor

Amy Morrison

Section Editor

Reviewer's Responses to Questions

**Key Review Criteria Required for Acceptance?**

**Methods**

-Are the objectives of the study clearly articulated with a clear testable hypothesis stated?

-Is the study design appropriate to address the stated objectives?

-Is the population clearly described and appropriate for the hypothesis being tested?

-Is the sample size sufficient to ensure adequate power to address the hypothesis being tested?

-Were correct statistical analysis used to support conclusions?

-Are there concerns about ethical or regulatory requirements being met?

Reviewer #1: There is a dearth in entomological data for most vector-borne diseases and the authors aim to fill that gap and to develop tools that may assist others to do so. They use the term “macroarea” but this is beyond the scope of their work and suggest refining. 

Objectives are ok, the text needs to be tightened. Ln 128 the data only enriches this small area within each country. State “our objectives are 1. Etc”

Differing collection methods were used in sites, some comment should be made on possible impact. Why were these sites selected? 

A scale is needed on the map. 

While data is collected on sites environment it is not really discussed anywhere in detail. 

Why were CDC bottle assays not completed in other places? This is a weakness in the data that should be discussed. 

Are the actual sand fly collections done inside homes? If if so was ethics obtained?

Reviewer #2: Methods used are suitable and the novel bioassay described here is very relevant for monitoring of insecticide resistance in sand flies. 

Some further comments:

1. Sampling areas, sand fly collection and sample handling

- Table 1: replace (X, Y) with Coordinates

- Figure 1: could the top map, showing locations in Greece and Turkey, be slightly readjusted so location TU4 is now almost off the map, whereas there is no need to show so much of the sea in the eastern part.

- Line 161: was obtained FROM… (not by)

- Are there any data on the number of human VL/CL cases in the areas where samples were collected?

5. Insecticide susceptibility bioassays

- It would be useful to provide a reason why the doses generally used for Aedes and Culex, rather than Anopheles, mosquitoes were used here.

- Please provide number of females tested per bottle (even if it is a range).

Section ii on individual glass vial bioassays: 

- It would be good to provide a picture of this novel method (especially if this novel bioassay could be pictures next to a standard CDC bottle). 

- Were the flies not kept for 24 hours to record mortality? If so, why not? Surely this would make it more comparable with the standard CDC bottle bioassay described in the previous section.

Reviewer #3: Please see Editorial and Data Presentation Modifications

**Results**

-Does the analysis presented match the analysis plan?

-Are the results clearly and completely presented?

-Are the figures (Tables, Images) of sufficient quality for clarity?

Reviewer #1: The species diversity is well presented – with different vectors in Greece compared to Turkey/Iran. Is there any link to disease burden that can be made with this? 

KDR is an interesting presentation, but here all species have grouped into one it would be good for places like Adana that individual species are shown. Although it is assumed the appendix is broken down into granular forms?

Unfortunately, bioassays were only done in Greece that had no real evidence of kdr, (would have been good to compare to Adana). While significantly reducing the deltamethrin concentration used in the assay, reduced the knockdown rate, that would be expected.

The individual assay is a good way to separate out species differences. How easy this could be applied in a vector control programme should be discussed. It was good to see that species ID of specimens were done, but it may have been interesting to include individual kdr here as well.

Reviewer #2: Results are clearly presented and described. 

Some further comments:

- Table 2: please add codes of sampling sites to their names (e.g. GR1 next to Attica/Kalentzi). Or add in the footnote which sites GR2-GR3 and TU2-TU3 were.

- Line 274: Heterozygosity in P. Bobbi specimen is mentioned. Please provide both codons - or did you mean this was a 1014F/S heterozygote?

- Line 281: heterozygosity in Iranian P. papatasi - please provide the codons observed.

- Figures 2 and 3: It should be :Knock down rate following deltamethrin exposure…” and not “…upon Deltamethrin…”.

- Were sand flies that were used in bioassays (either CDC or novel individual ones) then also genotyped? It would make sense to provide results giving both phenotype and genotype data as both are necessary to asses the insecticide resistance status - mutations by themselves, even if associated with resistance, do not necessarily mean the specimens are resistant. For example, in India it is known that P. argentines are resistant to DDT, have kdr mutations, yet still appear susceptible to pyrethroids. Provision of phenotypic and genotypic data is crucial for surveillance and control efforts, specially since leishmaniasis control largely depends on insecticides/vector control.

Reviewer #3: Please see Editorial and Data Presentation Modifications

**Conclusions**

-Are the conclusions supported by the data presented?

-Are the limitations of analysis clearly described?

-Do the authors discuss how these data can be helpful to advance our understanding of the topic under study?

-Is public health relevance addressed?

Reviewer #1: Overall, the conclusions are supported by the data, but in some instances this may need to be toned down a little. However, this is a public health issue globally and surveillance in this area needs to increase not just for sand flies but all vector-borne diseases. 

Ln 396 – 397 recall over time can be problematic this should be mentioned. 

401-404 – Bangladesh, and India have reported high kdr levels as referenced - however what has not been discussed is that these populations still remain susceptible to pyrethroids in bioassay that are currently in use for sand fly control and effective VL elimination (DOI: 10.1016/S1473-3099(24)00420-1 ) the low impact of kdr alone has also been observed in malaria transmitting mosquitoes. While similar correlations cannot be made with this data due to the lack of bioassay data combined with kdr it needs to be discussed here. 

417 – totally agree, and as the manuscript mentions the WHO sandfly assays it would be good to mention if those are higher, lower or same as the mosquito ones previously used. It is a good indicator as to how far off the current assays may be.

The individual bioassays are an interesting concept – although operationally in a programme may be challenging. Have any sample size calculations been attempted on the number sand lied needed to collect, test to get a viable data set?

Reviewer #2: Discussion and Conclusions are clear and supported by the data presented.

Some further comments:

- IR profiling tools are mentioned. Having a tool similar to IR Mapper for malaria vectors would be hugely beneficial for all those involved in leishmaniasis research, surveillance and control efforts. I am not suggesting that the authors should make such a tool but it would be good to see something like it mentioned in Discussion.

- Paragraph starting in line 381: these differences are indeed interesting. Surely there is a need for further research on bionomics of these sand fly species and populations (and others) to inform the control and surveillance efforts.

- Paragraph in lines 393-404: yes, regional IR surveillance is definitely needed, providing both phenotypic and genotypic data. But another crucial thing is lack of good control tools - and the fact that we clearly need to stop relying on the use o pyrethroids and start using other insecticides suitable for public health purposes.

- Line 429: if starting a sentence with a species name, spell the genus name out in full.

- In line with what is suggested in Conclusions (lines 465-468): To further improve both species identification, insecticide resistance genotyping and detection of pathogens, an approach based on high-throughput barcoding/amplicon seq might also be beneficial, especially as morphological identification of sand flies is much more complicated than that of mosquitoes.

Reviewer #3: Please see Editorial and Data Presentation Modifications

**Editorial and Data Presentation Modifications?**

Reviewer #1: It would be good to see kdr data where it exists presented by species as this is a consistent drive by the paper. 

The map could be clearer with appropriate scales etc.

Assume granular data will be available in appendices

Reviewer #2: Overall: language should be slightly revised, some sentences are awkward.

Reviewer #3: The article by Balaska et al reports interesting findings in a not well investigated field, insecticide resistance in sand flies. Moreover the paper proposes a protocol for testing in single sand flies, to overcome the impossibility to easily define sand fly species. In my opinion this is a relevant work, I have only minor revision to propose.

62 specify the geographic area, worldwide?

67 71 sentence not clear, please revise

174 please clarify if pcr was performed in single on every specimen

241 please specify the maximum time of the observation

213 reported in materials and methods

215 217 this sentence seems a materials an methods part

246 I suggest to insert more details on molecular identifications, as percentages of identity with deposed sequences and/or a three showing obtained clusters

table 2 I suggest to report also the number of sand flies and percentages between brackets 

333 the number of sand flies tested is unclear, please specify

**Summary and General Comments**

Reviewer #1: The paper delivers more data to the entomological surveillance community which is much needed. It highlights some of the critical gaps that need filing to make informed decisions on fighting vector-borne diseases. 

 While it would have been nice to see a correlation made between kdr and bioassay data, it is good to see that Greece has yet to develop a kdr issue. 

Please use place names in text and delete all codes, G1, G2 etc for a) consistency and b) readability

As colony sand flies not used, wild caught can be dropped

18 remove “up coming”

44-46- rephrase

49-key are capacity and resources

57 – how?

71-possible

88- suggest “WHO recently established”

107-108 – change to “hinder” rather than “deter”

120 – replace “include” – “are”

249-251- not easy to read

358 – care as reads 3 countries per region

357 – replace wild – “sand fly”

359-delete on-focus

414-does the sand fly population live in pyrethroid sprayed environment?

450 – “significantly” is a stretch.

Reviewer #2: (No Response)

Reviewer #3: (No Response)

PLOS authors have the option to publish the peer review history of their article (what does this mean?). If published, this will include your full peer review and any attached files.

Reviewer #1: No

Reviewer #2: No

Reviewer #3: No

Figure Files:

Data Requirements:

Reproducibility:

References

---

## [Editor Report · Decision Letter 1]

22 Oct 2024

Dear Prof. Vontas,

Thank you very much for submitting your manuscript "Development and application of species ID and insecticide resistance assays, for monitoring sand fly Leishmania vectors in the Mediterranean basin and in the Middle East" for consideration at PLOS Neglected Tropical Diseases. As with all papers reviewed by the journal, your manuscript was reviewed by members of the editorial board and by several independent reviewers. The reviewers appreciated the attention to an important topic. Based on the reviews, we are likely to accept this manuscript for publication, providing that you modify the manuscript according to the review recommendations. 

Sincerely,

Yara M. Traub-Csekö

Academic Editor

Amy Morrison

Section Editor

Figure Files:

Data Requirements:

Reproducibility:

References

---

## [Editor Report · Decision Letter 2]

8 Nov 2024

Dear Prof. Vontas,

We are pleased to inform you that your manuscript 'Development and application of species ID and insecticide resistance assays, for monitoring sand fly Leishmania vectors in the Mediterranean basin and in the Middle East' has been provisionally accepted for publication in PLOS Neglected Tropical Diseases.

Best regards,

Yara M. Traub-Csekö

Academic Editor

Amy Morrison

Section Editor

Shaden Kamhawi

co-Editor-in-Chief

Paul Brindley

co-Editor-in-Chief

---

## [Editor Report · Acceptance letter]

26 Nov 2024

Dear Prof. Vontas,

We are delighted to inform you that your manuscript, "Development and application of species ID and insecticide resistance assays, for monitoring sand fly *Leishmania* vectors in the Mediterranean basin and in the Middle East," has been formally accepted for publication in PLOS Neglected Tropical Diseases.

Best regards,

Shaden Kamhawi

co-Editor-in-Chief

Paul Brindley

co-Editor-in-Chief
